# Outcomes of Penta-Refractory Multiple Myeloma Patients Treated with or without BCMA-Directed Therapy

**DOI:** 10.3390/cancers15112891

**Published:** 2023-05-24

**Authors:** Shebli Atrash, Aytaj Mammadzadeh, Fulei Peng, Omar Alkharabsheh, Aimaz Afrough, Wei Cui, Zahra Mahmoudjafari, Al-Ola Abdallah, Hamza Hashmi

**Affiliations:** 1Levine Cancer Institute, Carolinas Healthcare System, Charlotte, NC 28204, USA; 2US Myeloma Research Innovations Research Collaborative (USMIRC), Westwood, KS 66205, USA; 3Division of Hematology/Oncology, Mayo Clinic, Rochester, MN 55905, USA; 4Department of Internal Medicine, Mercy St. Louis Hospital, St. Louis, MO 63141, USA; 5Division of Hematology/Oncology, The University of South Alabama Mitchell Cancer Institute, Mobile, AL 36604, USA; 6Division of Hematology/Oncology, UT Southwestern Medical Center, Dallas, TX 75390, USA; 7Department of Pathology & Laboratory Medicine, University of Kansas Medical Center, Kansas City, KS 66160, USA; 8Division of Pharmacy, University of Kansas Medical Center, Westwood, KS 66160, USA; 9Division of Hematologic Malignancies & Cellular Therapeutics, University of Kansas Medical Center, Westwood, KS 66160, USA; 10Division of Hematology/Oncology, Medical University of South Carolina, Charleston, SC 29425, USA

**Keywords:** plasma cell disorders, relapsed multiple myeloma, penta-class refractory myeloma, BCMA-directed therapy

## Abstract

**Simple Summary:**

Refractoriness to the five main myeloma treatments, including lenalidomide, pomalidomide, bortezomib, carfilzomib, and either daratumumab or isatuximab, define penta-refractory myeloma. Pent-refractory myeloma is a challenging disease that does not respond adequately to standard treatment approaches. However, clinical trials offer hope for innovative approaches. B-cell maturation antigen (BCMA) is a novel target for plasma cells. Initial clinical trials showed promising results in refractory myeloma. Nevertheless, measuring the benefits specifically for pent-refractory myeloma is required. In this retrospective analysis, we demonstrate that the BCMA-targeted approach has changed the outcomes of penta-refractory myeloma. While penta-refractory myeloma patients may benefit from BCMA-targeted therapy, more novel treatment options are needed to overcome its resistance.

**Abstract:**

Despite advances in treatment, outcomes remain poor for patients with penta-relapsed refractory multiple myeloma (RRMM). In this retrospective analysis, we evaluated the survival outcomes of penta-RRMM patients treated with (BCMA)- directed therapy (BDT). We identified 78 patients with penta-RRMM. Median age was 65 years, 29 (37%) had R-ISS stage III disease, 63 (81%) had high-risk cytogenetics, and 45 (58%) had extra-medullary disease. Median LOT prior to penta-refractory state was 5 (3–12). Amongst penta-RRMM, 43 (55%) were treated with BDT, 35 (45%) were not treated with BDT. Type of BDT received included belantamab mafadotin 15 (35%), Chimeric Antigen Receptor T-cell therapy 9 (21%), BCMA monoclonal antibody 6 (14%), and Bispecific T-cell engager 2 (5%). Eleven (25%) patients received more than one BDT. No significant differences were identified between baseline characteristics for the two groups. Patients treated with a BDT had better median overall survival, 17 vs. 6 months, HR 0.3 *p*-value < 0.001. Poor performance status, white race, and high-risk cytogenetics were associated with worse outcomes, whereas using a BDT was associated with better outcomes. Patients with penta-refractory MM have poor outcomes. Our retrospective analysis showed a significant survival benefit using BDT when compared to non-BDT for patients with penta-RRMM.

## 1. Introduction

Multiple myeloma is a hematological malignancy that leads to the accumulation of plasma cells in the bone marrow, resulting in anemia, renal failure, lytic bone lesions, or hypercalcemia. While there have been significant advancements in the treatment of myeloma over the years, with new drugs approved almost every year, it remains a challenging disease to manage [1]. One of the reasons for this challenge is that myeloma is a heterogeneous disease, genetically and clinically, meaning that it can present differently in different patients, making it challenging to develop a one-size-fits-all treatment approach. Additionally, myeloma is not curable and can develop resistance to treatment over time, further complicating the management of the disease and underscoring the need for more novel drugs. Therefore, researchers and clinicians continue to work hand in hand towards developing new and innovative therapies for myeloma, with the goal of improving outcomes for patients living with this disease.

Treatment options for MM have grown vastly in the last 25 years with the introduction of several new classes of drugs and a better understanding of the disease pathophysiology, and the development of new therapeutic agents such as proteasome inhibitors (PI), anti-CD38 monoclonal antibodies, and immunomodulatory drugs (IMiDs). In addition, novel agents such as B cell maturation antigen (BCMA)-targeting agents have improved outcomes for patients with MM [2,3].

The effectiveness of different treatments can vary greatly depending on various factors, such as the stage of the disease, the patient’s comorbidities, and myeloma genetics. Measuring progress allows clinicians to determine whether a particular treatment is working and adjust the treatment plan accordingly. This topic is particularly important in myeloma, as the disease can develop resistance to treatment over time, and patients may need to switch to different therapies to achieve optimal outcomes.

Additionally, researchers are monitoring long-term outcomes in patients with myeloma. As patients live longer with the disease, it is essential to track their progress to ensure that they receive appropriate and effective care.

The introduction of novel agents has significantly increased the depth of response as well as progression-free survival (PFS) and overall survival (OS) for patients with MM [4]. Despite significant gains with the introduction of novel agents, the care of relapsed refractory multiple myeloma (RRMM) remains challenging, and the recurrence of MM and disease progression is typical even after achieving deep remissions [5].

Penta-RRMM is defined as resistant to at least two IMiDs, i.e., lenalidomide and pomalidomide; two different Pis, i.e., bortezomib, ixazomib and/or carfilzomib; and one of the CD38 monoclonal antibodies, i.e., daratumumab or isatuximab. While most clinical trials report outcomes based on the number of previous lines of therapy, refractoriness to previous therapies is clinically more relevant, and penta-RRMM remains a therapeutic challenge [6].

BCMA is a receptor expressed on the cell surface of plasma cells and helps to induce B-cell proliferation, differentiation, and survival [7,8]. BCMA therapy comes in three main forms: antibody drug conjugates (ADC), CAR-T cell therapy, and bispecific T-cell engagers (BiTEs). These therapies have shown promising results in clinical trials, with high response rates and durable responses in patients with RRMM. However, ADC’s are not available as standard-of-care therapy in the US yet as it was voluntarily withdrawn for further testing.

CAR-T cell therapy involves collecting patients’ T cells and genetically modifying them to express a chimeric antigen receptor (CAR) that targets BCMA. These CAR-T cells are then infused back into the patient, where they can target and kill myeloma cells. CAR-T cell therapy has been shown to have high response rates in patients with RRMM, with some studies reporting overall response rates of over 80–97% [9].

BiTEs are another form of off-the-shelf BCMA therapy that works by binding to both BCMA on myeloma cells and CD3 on T-cells. BiTEs have also shown promising results in clinical trials, with some studies reporting overall response rates of over 60% [10].

The availability of BCMA therapy has transformed the treatment of myeloma, particularly for patients with relapsed or refractory disease who may have limited treatment options. These therapies have shown high response rates and durable responses in clinical trials, and they can potentially improve outcomes for patients with myeloma. BCMA-targeted therapies include antibody–drug conjugates (ADCs), bispecific T-cell engagers, monoclonal antibodies (MOA), and chimeric antigen receptor T-cell therapies (CAR-T) [10,11,12,13].

Clinical trials and real-world data are two different types of research used in myeloma to evaluate the safety and efficacy of different treatments. While both types of research are essential, they differ in several key ways.

Clinical trials that led to approved novel treatments for penta-refractory myeloma were tightly controlled studies with strict eligibility criteria designed to test the safety and effectiveness of new drugs or therapies [14]. However, real-world data come from a more diverse population of patients than clinical trials, and can provide valuable insights into how treatments work in real-world settings. Besides the level of control over the study population, patients in clinical trials typically require frequent monitoring to ensure that they are receiving the treatment as intended and to monitor for adverse events. Overall, both clinical trials and real-world data are important in myeloma research, and they provide complementary insights into the safety and effectiveness of different treatments. Clinical trials are essential for appraising the safety and efficacy of new therapies, while real-world data provide valuable insights into how treatments work in less controlled settings. While randomized controlled trials are the gold standard for evaluating the safety, efficacy, and approval of novel clinical interventions, such trials are not feasible in all situations. Hence, this retrospective analysis focuses on the outcomes of patients with penta-RRMM in the era of BCMA-targeting therapy.

## 2. Patients and Methods

This retrospective analysis was performed at the University of Kansas Medical Center (Westwood, KS, USA) in collaboration with the United States Myeloma Research Innovations Research Collaborative (USMIRC) between January 2015 to July 2022 for patients with penta-RRMM. The study received approval from the University of Kansas Institutional Review Board. Our inclusion criteria include patients with RRMM, who were identified as penta-refractory patients at the University of Kansas Health System that were defined as RRMM, refractory to all three classes including (1) PI (bortezomib and carfilzomib), (2) IMiDs (lenalidomide and pomalidomide), and (3) CD38 monoclonal antibodies (either daratumumab or isatuximab), as a single agent or in combination therapy. Our database identified seventy-eight patients that met the above criteria.

A progression within 60 days of a drug-containing regimen or a response less than the partial response (PR) defined the refractoriness of myeloma to a drug, as per previously published consensus criteria [15]. The time when patients met refractoriness criteria was defined as time zero (T0).

Baseline characteristics were obtained retrospectively from electronic medical records, including high-risk cytogenetics, disease stage, lines of therapy (LOT), treatment response, and survival outcomes. High-risk cytogenetics were defined as the presence of t(4;14), t(14;16), t(14;20), del 17p, or 1q21 gain or amplification.

Descriptive statistics were performed to summarize patient characteristics and disease status: a Fisher exact test and Mann–Whitney U test (Wilcoxon Rank Sum test) for categorical and continuous data, respectively. A survival analysis and multivariate analysis were performed. The Kaplan–Meier method was performed using the software R (Vienna, Austria) v2.15.1 and the survival package [16,17]. OS was defined as the time between T0 and the date of death. Patients without a recorded death date were censored for OS at their last contact date. BCMA-targeting agents were defined as either chimeric T-cells, antibody-drug conjugates, or bispecific T-cell engagers that used BCMA as a target.

## 3. Results

### 3.1. Patients Characteristics

The study reviewed a total of 78 consecutive patients with penta-RRMM. The patients’ median (range) age was 65 years (42–83) at T0; 51% were males. The median (95% confidence interval (CI)) estimated follow-up from T0 was 12 (5, 29) months. The baseline characteristics at T0 are shown in Table 1. The median number of prior LOT (range) before T0 was 5 (3–12); 67 (85%) had a prior stem cell transplant. Interestingly, most patients had high-risk cytogenetics (81%) and extramedullary disease (58%), as highlighted in Table 1. Post-T0, 43 patients (55%) received BCMA-targeting treatments—Figure 1. The type of BCMA-targeting treatments patients received included belantamab mafodotin in 24 (55%), Chimeric Antigen Receptor T cell therapy (CAR-T) in 17 (40%), BCMA monoclonal antibody in 12 (28%) [18], and Bispecific T-cell engager in 2 (5%). A few patients received more than one type of BCMA-targeted therapy—Figure 1. Eleven (25%) patients received more than one different BCMA-targeting therapy. The median time from the initial diagnosis to the penta-refractory state in the BCMA-treated versus (vs.) non-BCMA-treated patients were 66 (4–176) and 48 (14–135) months, respectively. The post-T0 BCMA- (*n* = 43) and non-BCMA-treated (*n* = 35) patients’ baseline characteristics were not statistically different except for performance status. The patients who received BCMA-targeted therapy were more likely to have a lower performance score of 1 vs. a PS of 2 in patients who did not receive a BCMA-targeted therapy—Table 1.

**Table 1 cancers-15-02891-t001:** Patient Characteristics with Penta-Class RRMM (*n* = 78).

Characteristics *n* (%)	All Patients (*n* = 78)	Received BCMA-Targeting Therapy (*n* = 43)	Did Not Receive BCMA-Targeting Therapy (*n* = 35)	*p*-Value
Male	44 (56%)	28 (65%)	19 (54%)	0.36
Age, years, median (range)	65 (42–83)	65 (42–83)	65 (44–83)	1
Race, no of patients (%)				
Caucasian	57 (73%)	30 (70%)	27(77%)	0.68
African American	18 (23%)	11 (26%)	7 (20%)	0.6
Hispanic	2 (3%)	2 (5%)	0	
Asian	1 (1%)	0	1 (3%)	
Performance status				
PS 1	48	31 (72%)	17 (48%)	0.04
PS 2	27	11 (26%)	16 (46%)	0.09
PS 3	3	1 (2%)	2 (6%)	0.58
MM paraprotein, no. of patients (%)				
IgG	44 (56%)	24 (56%)	20 (57%)	1
Non-IgG	21 (27%)	10 (23%)	11 (31%)	0.86
Light chain	13 (17%)	9 (21%)	4 (12%)	0.73
Baseline R-ISS stage, no of patients (%)				
Stage III	29 (37%)	11(26%)	18 (51%)	0.27
Stage II	26 (33%)	20 (47%)	6 (17%)	0.28
Stage I	18 (23%)	11 (26%)	7 (20%)	0.86
Unknown	5 (7%)	1 (2%)	4 (11%)	
Cytogenetics, no of patients (%)				
High-risk	63 (81%)	35 (81%)	28 (80%)	1
High-risk + 1q gain	66 (84%)	35 (81%)	31 (88%)	0.54
Standard risk	15 (19%)	8 (19%)	7 (20%)	1
Extramedullary disease	45 (58%)	25 (58%)	20 (57%)	1
Previous autologous SCT	67 (85%)	34 (81%)	32 (91%)	0.34
Median LOT prior to T0	5 (3–12)	5 (3–8)	6 (3–12)	

PS: performance status, R-ISS: Revised multiple myeloma international staging system, RRMM: relapsed/refractory multiple myeloma, SCT: stem cell transplant, LOT: lines of therapy.

In the subgroup of patients with extramedullary disease (EMD) 45 (58%) patients were identified. Diagnosis of EMD was based on biopsy (if applicable) and imaging using computed tomography (CT) or 18F-FDG PET. For those who had CNS involvement, we used both imaging studies that included either CT scan or magnetic resonance imaging (MRI) and CSF analysis that confirmed elevated monoclonal plasma cells. Thirty-Six (46%) patients had plasmacytoma that involves muscle, skin, soft tissue, liver, and pleural fluid, while nine (12%) had CNS involvement.

Patients with high-risk cytogenetics were evaluated in this group. Patients with double-hit cytogenetics (more than one high-risk cytogenetic factor) were observed in 26 (41%) patients. The total median LOT till the last office visit/death was eight (3–15) lines, while the median LOT prior onset of penta-RRMM was five (3–12) lines. Median time from diagnosis of myeloma till the onset of penta-RRMM was 49 (4–176) months, while the median time from onset of penta-RRMM to last office visit/death was 11 (1–67) months.

### 3.2. Survival Outcomes

After a median follow-up (interquartile range (IQR)) of 12 months (5, 29), the median (95% CI) OS for the entire cohort was 12 months (9, 17) from T0 (Figure 2). The median OS for patients treated with a BCMA-targeting agent compared with patients who did not receive a BCMA-targeting agent was 17 vs. 6 months, respectively (HR 0.3, 95% CI (0.18, 0.51), *p* < 0.0001). In the multivariate analysis, poor performance status, high-risk cytogenetics, and white race were associated with worse outcomes, whereas the use of a BCMA-targeting agent was associated with better outcomes—Figure 3 and Figure 4. Out of 78 patients, 17 received BCMA CAR T and reported a median OS of 29 months (95% CI 17, not reached (NR)), and 24 received belantamab mafodotin with a median OS of 23 months (95% CI 16, NR).

Up to date, 61 (78%) of patients from all patients died, and the commonest cause of death was progressive disease in 54 (89%) patients; other causes reported were the following: sepsis secondary to treatment in 2 (3.2%) patients, AML/MDS in 2 (3.2%) patients, acute coronary disease/congestive heart failure in 2 (3.2%) patients, and lung cancer in 1 (1.6%) patient.

## 4. Discussion

This study highlights the poor outcome for patients with RRMM, who have become refractory to all five drugs from three major classes of medications currently used in clinical practice for RRMM. These patients need access to newer classes of drugs with different mechanisms of action. Our study defines a new benchmark for contemporary expectations for survival. In addition, our study underscored a sizable proportion of patients with high-risk cytogenetics and extramedullary disease, reflecting the current expectations of an increased percentage of high-risk disease features with each relapse.

Despite the similarity between triple-class refractory and penta-refractory myeloma, the latter conveys extra concerns about T-cell fitness due to multiple previous LOTs. The international myeloma working group (IMWG) study set the benchmark for triple-class refractory disease by interrogating data from existing medical records from multiple centers across North America, Europe, and Asia-Pacific. The study enrolled 543 patients, and refractoriness to IMiDs and PIs predicted a median PFS of 5 months and a median OS of 15.2 months [19]. Additionally, the penta-refractory disease was associated with guarded outcomes. In another retrospective analysis of RRMM patients from 14 academic institutions in the United States, the survival analysis of penta-refractory (refractory to one CD38 monoclonal antibody + two PIs + two IMiDs) patients revealed a median OS of only 5.6 months (3.5–7.8). Response rates declined with each subsequent regimen, reaching 18% with the fifth line of therapy [6].

The treatment of RRMM remains a challenge despite the increasing number of available therapies, with a shorter survival time with increasing refractoriness. Current treatment approaches for triple refractory MM include reusing previous treatment regimens in different combinations, conventional chemotherapy, salvage ASCT, selinexor, belantamab, melflufen, chimeric antigen receptor (CAR) T-cell, and bispecific T-cell engager technology. However, the OCEAN study showed concerns about the mortality rate using Melflufen in patients with RRMM [20].

On the other hand, BCMA-targeting therapies have changed the outcomes of penta-refractory myeloma. For example, despite failing to show superiority against pomalidomide in the DREAMM-3 study [21], retrospective analysis for penta-refractory myeloma patients treated with belantamab mafodotin showed a median PFS of 4.3 months and median OS of 10.7 months [22]. Recently, two separate retrospective studies of real-life experience, with each having enrolled around 30 belantamab-mafodotin-treated patients, reported an OS of 8 months with triple class-refractory myeloma and 6.5 months in penta-RRMM [23,24]. These outcomes were similar to those reported in the DREAMM-2 trial [13].

In addition, in the US, CAR T-cell therapy was approved for RRMM with favorable overall response rates between 73 and 98% after four prior LOTs [11,12]. The median PFS for idecabtagene vicleucel was 8.8 months, while the median PFS has not yet been ascertained for ciltacabtagene autoleucel [11]. A retrospective multicenter CAR T consortium also saw an almost identical PFS of 8.9 months [25]. The CAR T consortium reported the PFS outcomes of 41% of penta-refractory patients, demonstrating an HR of 0.93 (95% CI 0.46, 1.87, *p*-value 0.15) [25]. Additionally, the bispecific antibody teclistamab induced a response rate of around 63% in the MajesTEC-1 clinical trial for RRMM after four prior LOTs [10]. In the MajesTEC-1 trial, 30 out of 50 patients with penta-RRMM reported an overall response [10]. Despite concerns about T-cell fitness and an immunosuppressive microenvironment in the penta-refractory population, these agents demonstrated a clear clinical benefit [26]. Recently, data from MajesTEC-1 were compared to the nationwide de-identified electronic health record-derived Flatiron Health MM cohort database for patients with triple-class exposed RRMM. MajesTEC-1 produced a statistically better progression-free survival with a HR of 0.43 [95% CI: 0.33–0.56]; *p* < 0.0001 [27].

Several other bispecifics that bind BCMA and CD3 to redirect T cells to myeloma cells are being developed and are presenting phase 1 results. (Table 2). Bispecifics targeting antigens beyond BCMA, such as GPRC5D (GPRC5DxCD3, talquetamab, and RG6234) and FcRH5 (FcRH5xCD3, cevostamab), are in clinical development showing promising results for heavily treated myeloma.

The most advanced of those novel targets is the GPRC5D. It is considered an important addition to the myeloma armamentarium. While BCMA is expressed on all mature plasma cells, GPRC5D is selectively expressed on the surface of myeloma cells, making it an attractive target for immunotherapy against abnormal plasma cells. Several studies have investigated the potential of GPRC5D-targeted immunotherapy in the treatment of RRMM. One approach involves using a bispecific antibody against CD3 and GPRC5D, redirecting T cells to mediate the killing of GPRC5D-expressing myeloma cells. Talquetamab is a bispecific antibody tested in a phase 1 clinical trial that enrolled 232 patients; 102 were treated intravenously and 130 subcutaneously [28]. The trial suggested two recommended phase 2 doses (RP2D): 405 μg per kilogram weekly and 800 μg per kilogram every other week. The most common adverse events at RP2D were skin-related events, dysgeusia, and low-grade cytokine-release-syndrome events. Sixteen percent of patients had high-risk cytogenetics. For weekly dosing, at a median follow-up of 11.7 months, response rates were 70%, with a median duration of response of 10.2 months. For biweekly dosing, at a median follow-up of 4.2 months, the overall response rate was 64%, with a duration of response of 7.8 months. While 77% of patients were penta-exposed, only 25% of patients were penta-refractory. Of note, out of sixteen patients with BCMA treatment exposure, eight patients (50%) had a response.

A different strategy for addressing GPRC5D involves using CAR-T cells specifically targeting this receptor, such as MCARH109. In a phase 1 dose-escalation study, patients with RRMM received MCARH109 at four different dose levels (25 × 10^6^, 50 × 10^6^, 150 × 10^6^, and 450 × 10^6^). The study allowed prior BCMA CAR T-cell therapy. The primary objective was to assess the safety of MCARH109 [29]. In the studied patient population, all individuals had been penta-exposed, with 59% having received prior BCMA therapy. The dose of 150 × 10^6^ CAR T cells was the maximum tolerated. The overall response rate was 71%. The median duration of response was 7.8 months. Because GPRC5D is present in the skin, taste buds, and nail beds, some on-target, off-tumor toxic effects were observed, such as rash, dysgeusia, and nail changes. However, when compared to talquetamab, MCARH109 induced a lower frequency and severity of skin rash and dysgeusia. In addition, the nail changes associated with GPRC5D-targeted therapies were reversible.

FcRH5, a type I membrane protein expressed on B cells and plasma cells, is the next target in development for penta-refractory myeloma. Cevostamab, a bispecific monoclonal antibody that engages both FcRH5 and CD3 on T cells, enables the T-cell-mediated killing of plasma cells. In a phase I trial (NCT03275103), heavily pre-treated RRMM patients received cevostamab Q3W for up to 17 cycles. Of the 16 patients completing the treatment, 81% were triple-class refractory, and 68% were penta-refractory. Notably, 31% had prior BCMA-targeted therapy, with 25% refractory to anti-BCMA treatment. Ten patients achieved ≥ complete response (CR), and five patients achieved a very good partial response (VGPR). In addition, most patients remained in remission beyond 6 and 12 months. However, as expected, treatment was associated with an increased risk of infections [30].

Our multivariate subset analysis could underscore many observations. Perhaps, the most important was the observation that more LOTs appeared to be trending toward better outcomes. Once again, prior LOT did not reflect this population’s unmet needs. Patients who have received multiple lines of therapy may have more refractory and aggressive disease, which can make it more difficult to deliver effective subsequent therapies. In addition, prior therapy may result in more treatment-related toxicities. On the other hand, more lines of therapy could also mean different a cancer biology with slower growth or more responsive disease, particularly if previous lines of therapies consist of recycling the same drugs with different combinations. We believe that the refractory status of myeloma is more prognostic than the number of prior lines of therapy. For example, patients who progress after quadruplet therapy, in the setting of front-line therapy with the GRIFFIN regimen that consists of daratumumab, lenalidomide, bortezomib, and dexamethasone [31], will be penta-refractory after progression to carfilzomib, pomalidomide, and dexamethasone in the second line. These patients have an unmet need to access novel therapies. However, such patients will not have access to novel therapies because those therapies are usually approved after four prior lines of therapy, such as CAR-Ts or teclistamab. Hence, designing clinical trials based on the prior treatments’ refractory status seems more representative of a homogenous population. In line with our own findings, Costa et al. demonstrated that when classifying patients with RRMM undergoing current therapy, drug-class refractoriness is a more effective measure than the number of prior lines of therapy. This research aligns with our understanding that resistance to specific drug classes is crucial in determining treatment response. By designing trials focusing on drug class refractoriness, we can better tailor therapies and develop more efficacious treatment strategies for patients with RRMM [32]. Costa et al. conducted a retrospective analysis on patients commencing new standard-of-care regimens for RRMM between 2016 and 2022. They assessed refractoriness to the three main classes of anti-myeloma drugs (PI, IMiDs, and Anti-CD38). Additionally, investigators categorized prior therapy lines into three groups: one line, two to three, or ≥four. The study found that median progression-free survival (PFS) was 16.0 months for the one-prior-line group, 11.0 months for those with two to three lines, and 6.1 months for patients with ≥four prior lines of therapy. Regarding drug class refractoriness, median PFS was 17.8 months for patients without or with only single-class refractoriness, 7.2 months for double-class refractory patients, and three months for triple-class refractory patients. Multivariable analysis revealed that the number of prior lines of therapy was not associated with PFS (*p* = 0.16), whereas drug-class refractoriness strongly predicted PFS (*p* < 0.001). Therefore, the authors concluded that drug class refractoriness is a superior predictor of response rates and median PFS compared to the number of prior lines of therapy [32].

**Table 2 cancers-15-02891-t002:** T-cell engagers underdevelopment for relapsed refractory multiple myeloma.

	Target	N	LOT	TCRRMM	PCRRMM	RR (All Doses)	RR (Rec Dose)	mPFS (Months)	mDOR (Months)
ABBV-383 [33]	BCMA	124	5 (3–15)	82%	35%	59%	57%	10.4 (5.0–19.2)	NR (12.9–NR)
Linvoseltamab * [34]	BCMA	252	5 (1–16)	81%	37%	50%	64%	n/a	n/a
Elrantamab * [35]	BCMA	55	5 (2–14)	91%	n/a	64%	64%	12m: 60%	17.1 (10.6–NE)
Teclistimab [10]	BCMA	165	5 (2–14)	78%	30%	63%	n/a	11.3 (8.8–1.1)	18.4 (14.9–NE)
HPN-217 * [36]	BCMA	62	6 (2–19)	76%	42%	57%	77%	n/a	12 (6–NE) *
Alnuctamab * [37]	BCMA	68	4 (3–11)	63%	28%	53%	65%	n/a	n/a
Talquetamab [28]	GPRC5D	288	5 (2–13)	72%	26%	73%	73%	7.5 (5.7–9.2)	9.3 (6.6–20)
RG6234 [38]	GPRC5D	107	5 (2–15)	67%	39%	67%	67%	n/a	10.5/12.5 month
Cevostamab [30]	FcRH5	53	6 (2–15)	72%	45%	53%	61%	n/a	n/a

TCRRMM: triple-refractory RRMM; PCRRMM: penta-refractory RRMM. RR: response rates. * data from oral presentations. NR: not reached. NE: not evaluable, n/a: not available.

In addition, our multivariate analysis showed that patients had similar outcomes regardless of extramedullary disease. This outcome could be partly due to the small sample size. However, the BCMA-targeting approach remained influential despite the high percentage of extramedullary myeloma (58%).

Additionally, the Caucasian race was associated with worse overall survival despite a similar frequency of high-risk cytogenetics and baseline performance status. A proportion of 10 out of 18 AA patients (56%) had high-risk cytogenetics vs. 27 out of 52 Caucasian patients (52%) (*p*-value 0.59), and 12 out of 18 AA patients (67%) vs. 34 out of 57 Caucasian (60%) had a PS of 0–1 (*p*-value 0.78). The worse outcomes in Caucasian patients could be due to the low sample size; nevertheless, it warrants further investigation. It is hard to draw conclusions from this observation due to the small number of patients included in this analysis.

Finally, as expected, a lower performance status score was associated with a better OS. However, patients with a performance status of 1 were more likely to receive a BCMA-targeted therapy—Table 1. Therefore, performance status was another confounding factor. In our multivariate analysis, treatment with BCMA-targeted drugs remained strongly positive despite adjusting for performance status.

Our study has several limitations. Firstly, this is a single-center retrospective study with relatively few patients. However, to the best of our knowledge, this is the largest “real-life” cohort of penta-RRMM patients adding valuable data to the existing literature in this area of unmet need. Secondly, this study includes highly heterogeneous treatments for patients with variable access to novel salvage options with clinical trials.

In addition, there is inevitable bias in a study such as this, primarily including data from a referral MM center. This is evident with the younger median age of the study population, reflecting both a referral bias and the fact that younger patients had better access to multiple BCMA-targeting therapy options. Thirdly, differences in survival between different BCMA-targeting therapies usually attributed to the unique mechanism of action (CAR T vs. antibody-drug conjugates) and patient characteristics (disease burden and comorbidities) could not be effectively studied in our analysis due to a small sample size and heterogeneous population.

## 5. Conclusions

Considering how rapidly the treatment of MM is evolving, soon, with the introduction of quadruplet regimens for newly diagnosed myeloma, most patients will be exposed in their earlier lines of therapy to PIs, IMiDs, and CD38 monoclonal antibodies. Therefore, treatment guidelines for the penta-refractory disease are needed. In addition, this study underscores the importance of reporting the refractory disease status in clinical trials in addition to/ instead of the numbers of prior lines of therapy. BCMA-targeting drugs with novel mechanisms of action are needed and can improve the outcomes of patients with penta-RRMM.

## Figures and Tables

**Figure 1 cancers-15-02891-f001:**
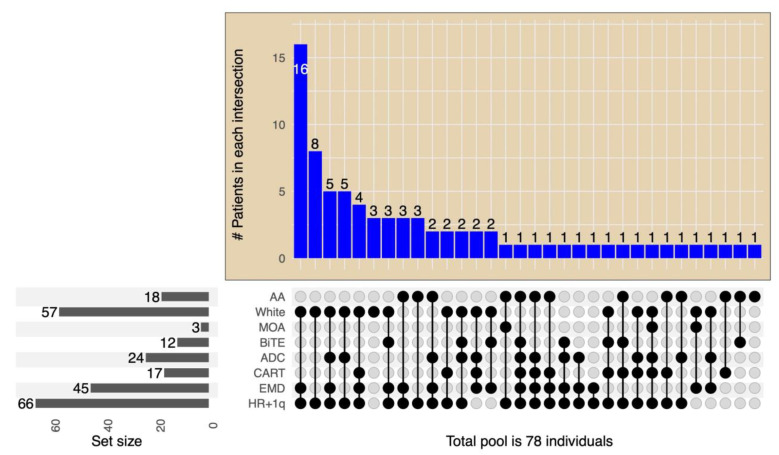
Chart of the size of each subset population based on treatment allocation (BCMA, ADC, CART) and disease characteristics (HR + 1q, EMD). On the left lower panel, bars represent the total number of each disease characteristic and BDT treatment subset. The intersection size is on the top and presented as columns. The intersection details for each bar are below it, with associations shown as dots–lines. HR + 1q: high-risk cytogenetics of 17p del, t(4;14), t(14;16), t(4,20), and gain of 1q. ADC: antibody-drug conjugate. MOA: monoclonal antibody. BiTE: bispecific T-cell engager. White: Caucasian race, AA: African American race, including CAR-T, ADC, or T-cell engagers.

**Figure 2 cancers-15-02891-f002:**
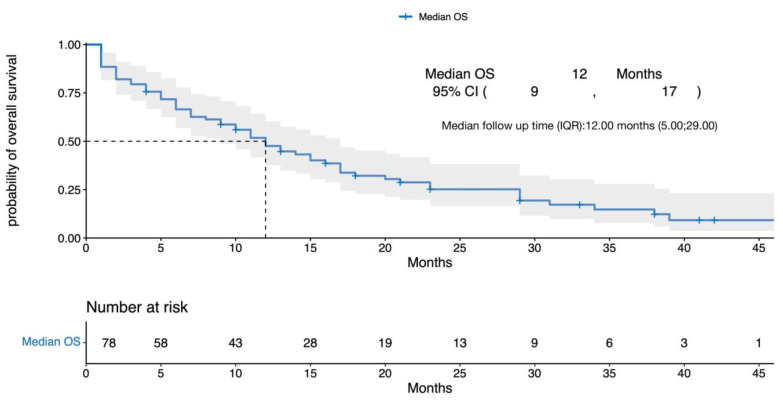
Kaplan–Meier curves for the overall survival of the entire population. OS = overall survival, CI = confidence interval, IQR = interquartile range. Gray area represents the 95% confidence interval for overall survival.

**Figure 3 cancers-15-02891-f003:**
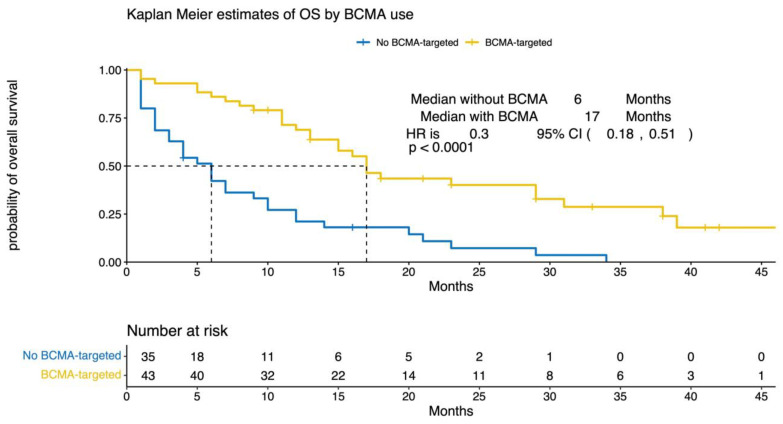
Clinical outcomes based on salvage treatment with BCMA-targeted therapy. Overall survival of penta-refractory/relapsed multiple myeloma patients who received BCMA-targeted therapy was compared to that of subjects who did not. The number of censored subjects at risk is also reported. A *p* < 0.05 was considered statistically significant. OS = overall survival, BCMA = B-cell maturation antigen, HR = hazard ratio, CI = confidence interval.

**Figure 4 cancers-15-02891-f004:**
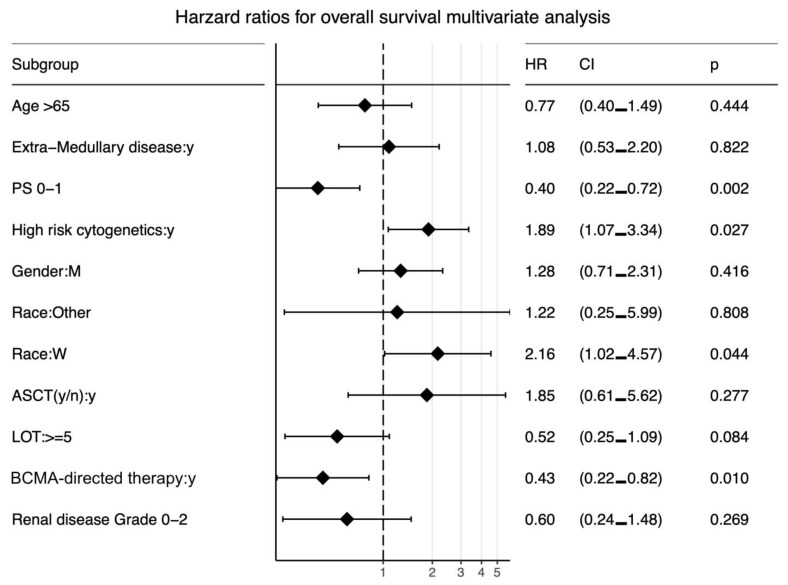
Forest plot for the hazard ratio for overall survival. PS: performance score, CI: confidence interval, HR: hazard ratio, LOT: lines of therapy, W: white, M: male. Higher HR for progression correlates with worse survival. Races were compared to African American race.

## Data Availability

The data that support the findings of this study are available on request from author, AA. The data are not publicly available due to local IRB restrictions.

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
