# Peer review of "Outcomes of Penta-Refractory Multiple Myeloma Patients Treated with or without BCMA-Directed Therapy"

_cancers, 2023, doi:10.3390/cancers15112891_

Round 1

Reviewer 1 Report

Minor issues should be considered:

-Abbreviations:

*Introduction, line 1. Multiple myeloma (MM). Abbreviation should be used thereafter: second and fifth paragraphs.

*RRMM is described on fifth paragraph. Abbreviation should be used thereafter: seventh paragraph.

*Patient and methods, last paragraph: T0, please, describe it.

*Discussion: Please, use abbreviations previously described (PIs,IMiDs, LOT, PFS, …).

-Typos in Discussion: after, described, multivariate, …

-Duplicate text: Information about the three main forms of anti-BCMA therapy are duplicated in paragraphs seventh and tenth.

-Table 2 does not read well, it appears misconfigured. Please, arrange it.

-Style: references' format should be adapted to Cancers.

Author Response

Thank you for your feedback

  • We fixed the abbreviations for PI, IMiD, MM, and RRMM.
  • We added a better description of T0 to the methods section
  • We fixed typos all over the manuscript.
  • Removed paragraph ten
  • We fixed Table 2
  • We changed references to cancers.

Reviewer 2 Report

This retrospective analysis focuses on the outcomes of patients with penta-RRMM in the era of BCMA targeting therapy.

The authors did not indicate :

- the chosen cut-off value del 17p positivity?

-which treatment received patients who did not received anti bcma?

In the discussion, there is a review of new drugs from paragraph 5 to 10. What is the point?

Paragraph 11 is very long.

Data are not really compared to their own results.

 There are a lot of spelling mistake « Simillar » « multivariat analysis » « from this obsercation ».

 Table 2 (T-cell engagers underdevelopment for relapsed refractory multiple myeloma) is not readable.

Author Response

Authors' answers:

  • FISH was used to detect del 17p. The cutoff was 5%.
  • The treatment for patients who did not receive anti-BCMA was highly heterogenous and consisted of recycling previously used chemotherapy agents with different combinations. Therefore, we focus on overall survival instead of progression-free survival.
  • We wanted to perform a quick review of the literature on immunotherapies. Typos were fixed throughout the manuscripts. We removed paragraph 10 to shorten the discussion.
  • We fixed Table 2

Reviewer 3 Report

Very importent issue, and it appears clearly that penta-refractory patients have bad outcomes.

Introduction:

I dont like the describtion of MM as a blood cancer. In first paragraph you mention MM as mostly not curable - i find MM incurable! Later in first paragraph: the ultimate gold in my opinion is CURE.

The introduction is very long, consider shorten it. The paragraph starting with "Additionally, monitoring...", consider whether it can be omitted.

2.Patiens and Methods

You describe 3 classes of refractory, put number in front of all three!

Your database identified 43 patients - later you have 78 patients.

3.Results

Consider to leave out: "A few pateints received more than one type of BCMA targeted therapies" - it is further described in the later sentence. 

4.Discussion

Very long.  I lack a discussion about treatment with BCMA directed therapy being far superior (17 vs 6 mo). Maybe also a economical aspect, as especially, CAR-T is extremely expensive. 

Author Response

Thank you for your feedback.

  • Changed blood cancer to hematological malignancy.
  • We removed "mostly" and "ultimate."
  • Added numbers for all three classes.
  • The total number of penta-refractory is 78. The manuscript was modified to reflect that.
  • This sentence was requested by another reviewer. We agree with this comment. But we decided to leave this sentence to help the reader understand the difference in the denominator.
  • We wanted to supplement this manuscript with a quick literature summary based on the journal request. Paragraph ten was removed to shorten the discussion.

Round 2

Reviewer 2 Report

no comment